# TASK-ORIENTED SEQUENTIAL GROUNDING IN 3D SCENES

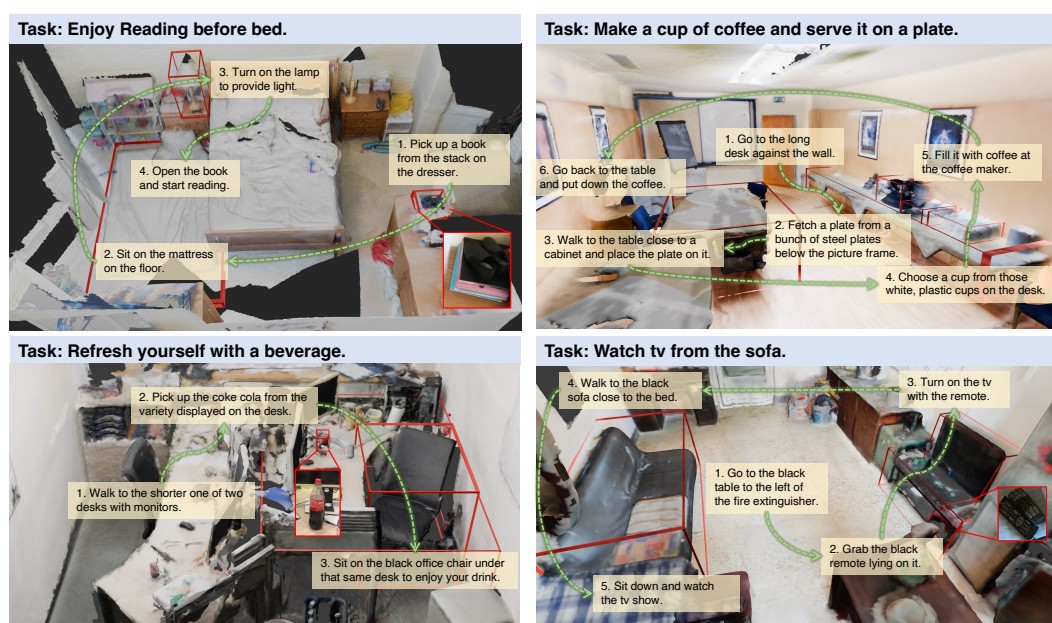

Figure 1: **The task-oriented sequential grounding task in 3D scenes (SG3D)**, wherein an agent is required to locate a sequence of target objects for detailed steps in a plan to complete daily activities. To solve this task, an agent must understand each step *in the context of the whole plan* to identify the target object, since a single step alone can be insufficient to distinguish the target from other objects of the same category. Additional resources can be found at sg-3d.github.io.

## ABSTRACT

Grounding natural language in physical 3D environments is essential for the advancement of embodied artificial intelligence. Current datasets and models for 3D visual grounding predominantly focus on identifying and localizing objects from static, object-centric descriptions. These approaches do not adequately address the dynamic and sequential nature of task-oriented grounding necessary for practical applications. In this work, we propose a new task: Task-oriented Sequential Grounding in 3D scenes, wherein an agent must follow detailed step-by-step instructions to complete daily activities by locating a sequence of target objects in indoor scenes. To facilitate this task, we introduce SG3D, a large-scale dataset containing *22,346 tasks with 112,236 steps across 4,895 real-world 3D scenes*. The dataset is constructed using a combination of RGB-D scans from various 3D scene datasets and an automated task generation pipeline, followed by human verification for quality assurance. We adapted three state-of-the-art 3D visual grounding models to the sequential grounding task and evaluated their performance on SG3D. Our results reveal that while these models perform well on traditional benchmarks, they face significant challenges with task-oriented sequential grounding, underscoring the need for further research in this area.

## 1 INTRODUCTION

Grounding natural language in the physical 3D world is crucial for advancing embodied artificial intelligence (Embodied AI) (Duan et al., 2022; Wang et al., 2023a), where robots must follow human instructions to complete complex tasks. Recent years have witnessed the collection of various datasets (Jia et al., 2024; Chen et al., 2020; Achlioptas et al., 2020; Zhang et al., 2023; Wang et al., 2023a; Kato et al., 2023) aimed at training and testing robust visual grounding models in 3D scenes (Zhu et al., 2023; 2024; Chen et al., 2022b; Guo et al., 2023; Jain et al., 2022). While these datasets have driven progress in 3D visual grounding, they largely borrow practices from 2D visual grounding, concentrating on identifying and localizing objects based on *object-centric* descriptions (Chen et al., 2020; Achlioptas et al., 2020). Such descriptions distinguish the target object from other objects by detailing its attributes and spatial relationships. However, this object-centric style may be insufficient for the embodied agent's application scenarios, where the language used to interact with agents often involves task assignments rather than mere object identification, as exemplified in SayCan (Ahn et al., 2022) and SayPlan (Rana et al., 2023). Thus, a significant yet overlooked gap exists between existing 3D visual grounding approaches and the task-oriented language demands of embodied agents. This disparity is highlighted in Fig. 2, which compares object-centric and task-driven visual grounding in 3D scenes.

To close this gap, we propose a new task: Task-oriented Sequential Grounding in 3D scenes. In this task, an agent is asked to accomplish a daily activity with a detailed plan in an indoor scene, by sequentially localizing one target object for each step. To solve this task, an agent must understand each step *in the context of the whole plan* to identify the target object, since a single step alone can be insufficient to distinguish the target from other objects of the same category.

To address this challenge, we constructed a large-scale dataset named SG3D. We compiled a set of RGB-D scans of realistic indoor scenes sourced from various 3D scene datasets, including ScanNet (Rozenberszki et al., 2022), ARKitScenes (Baruch et al., 2021), 3RScan (Wald et al., 2019), etc. These scenes encompass a variety of room types, such as bedrooms, kitchens, offices, bathrooms, and living rooms. We represent these scenes using 3D scene graphs (Armeni et al., 2019; Wald et al., 2020) derived from SceneVerse (Jia et al., 2024), which describe the objects' categories, attributes, and spatial relations within the scenes.

We further designed an automated generation pipeline that utilizes these scene graphs and GPT-4 (Achiam et al., 2023) to create diverse, high-quality daily tasks. Each task comprises a high-level description and a detailed plan, with the target object annotated for each step. To ensure the validity of the generated tasks, we conducted a human verification process to check if the tasks were appropriate for the scenes, if the plans were sufficient to accomplish the tasks, and if the target objects were correctly identified for each step. Invalid tasks were either filtered out or manually refined. Ultimately, the proposed SG3D includes *22,346 tasks* with *112,236 steps* across *4,895 real-world 3D scenes*, as exemplified in Fig. 1. Tab. 1 compares SG3D with existing 3D visual grounding benchmarks.

In our experiments, we adapted three state-of-the-art 3D visual grounding models to the sequential grounding task and evaluated them on SG3D. The models included 3D-VisTA (Zhu et al., 2023), PQ3D (Zhu et al., 2024), and LEO (Huang et al., 2023a). The results indicate that although these models excel on previous benchmarks, they struggle with the more complex and realistic grounding presented in the SG3D benchmark. This highlights the need for further research and development to improve performance in task-oriented sequential grounding scenarios for Embodied AI.

Our contributions are summarized as follows:

- We proposed a new task, Task-oriented Sequential Grounding in 3D scenes, to address the gap between object-centric and task-driven grounding required for practical Embodied AI applications.

- We constructed a large-scale dataset for this novel task, SG3D, which contains 22,346 tasks with 112,236 steps across 4,895 real-world 3D scenes.

- We adapted three state-of-the-art 3D visual grounding models (3D-VisTA, PQ3D, and LEO) to the sequential grounding task and evaluated them on SG3D. Experimental results indicate that these models struggle with task-oriented sequential grounding, highlighting the need for further advancements in this area.

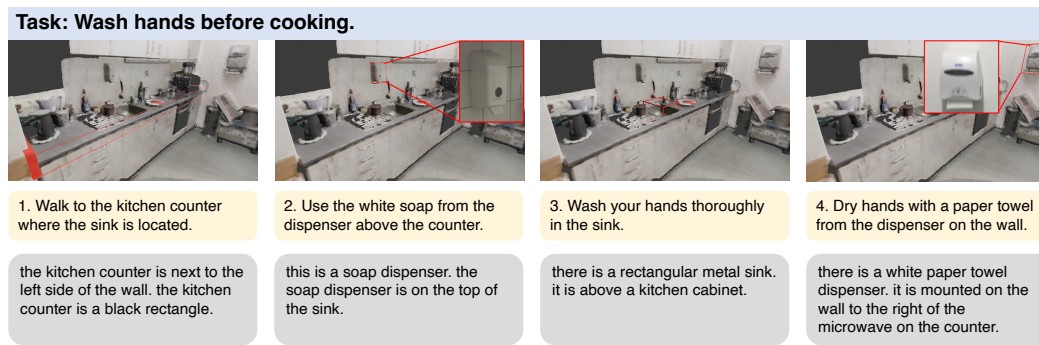

Figure 2: The comparison between task-oriented steps in SG3D (first row) and object-centric referrals in ScanRefer (second row) for the same target objects. Particularly, in step 3, the ScanRefer annotation describe the sink's shape, material, and spatial relation to the cabinet to identify it, while the corresponding step in SG3D avoids such details. The *context* provided by the task makes it easy to infer that the sink is near the soap dispenser mentioned in the previous step.

## 2 RELATED WORK

Table 1: **The comparison of SG3D with existing 3D visual grounding benchmarks.** SG3D expands the data scale of prior work by order of magnitude. "VG" stands for Visual Grounding, "SG" for Sequential Grounding, and and "MT" for Multiple Tasks. * Only new data is counted.

| Dataset | Task | Referral type | Text Source | Quality Check | Scene | Obj. | Avg. Text Len. | Vocab. | Total |
|---|---|---|---|---|---|---|---|---|---|
| ScanRefer (Chen et al., 2020) | VG | Object-centric | Human | ✓ | 1.5K | 33K | 20.3 | 4,197 | 52K |
| Nr3D (Achlioptas et al., 2020) | VG | Object-centric | Human | ✓ | 1.5K | 33K | 11.5 | 2,986 | 42K |
| Sr3D (Achlioptas et al., 2020) | VG | Object-centric | Template | ✓ | 1.5K | 33K | 9.7 | 158 | 84K |
| Multi3DRefer* (Zhang et al., 2023) | VG | Object-centric | Template w/ Rephrasing | ✓ | 1.5K | 33K | 15.1 | 7,077 | 20K |
| SceneVerse* (Jia et al., 2024) | MT | Object-centric | Human + GPT-3.5 | ✓ | 68K | 1.5M | 14.7 | 24,304 | 2.2M |
| **SG3D** | SG | Task-oriented | GPT-4 | ✓ | **4.9K** | **123K** | **70.5** | **8,136** | **22K / 112K** |

**3D Vision-Language**   3D vision-language (3D-VL) learning aims to bridge natural language and the 3D physical world (Zhu et al., 2023; 2024; Kerr et al., 2023), enabling embodied agents to comprehend their environment and communicate with humans effectively (Zhu et al., 2023; Rana et al., 2023). This emerging domain has established benchmarks for various tasks, such as visual grounding (Chen et al., 2020; Achlioptas et al., 2020; Abdelreheem et al., 2024; Zhang et al., 2023; Kato et al., 2023), question answering (Azuma et al., 2022; Zhao et al., 2022; Ma et al., 2023), and dense captioning (Chen et al., 2021). Beside many methods tackling single tasks (Guo et al., 2023; Wu et al., 2023a; Luo et al., 2022; Jain et al., 2022; Zhao et al., 2021; Chen et al., 2022b), unified models (Zhu et al., 2023; 2024; Chen et al., 2023c) and open-vocabulary approaches (Peng et al., 2023; Ding et al., 2023; Takmaz et al., 2023) have gained traction in recent literature. However, existing 3D visual grounding benchmarks primarily address *object-centric*, single-step grounding tasks, whereas realistic grounding sentences are typically driven by *task-related* context (Deng et al., 2024). In contrast to previous work, SG3D provides more natural and informative language and introduces diverse *contextual* information, as shown in Fig. 2.

**Grounded Task Planning**   Embodied AI focuses on the capabilities of agents to reason, plan, navigate, and act in 3D environments (Deitke et al., 2022; Huang et al., 2023a; Ahn et al., 2022). Grounded task planning is crucial as it enables these agents to execute human instructions effectively (Lin et al., 2023; Zhao et al., 2024). Benchmarks such as ALFRED (Shridhar et al., 2020), SAYPLAN (Rana et al., 2023), BEHAVIOR-1K (Li et al., 2023a), and TaPA (Wu et al., 2023b) evaluate these abilities by measuring the success of the agents' overall task plans. Others, like LoTA-BENCH (Choi et al., 2024), EgoPlan-Bench (Chen et al., 2023b), and G-PlanET (Lin et al., 2023), assess performance per step using rule-based or closed-set evaluations. Both specialized models (Yang et al., 2023; Zhang et al., 2022; Shridhar et al., 2020) and foundation models (Song et al., 2023; Wei et al., 2022; Li et al., 2022; Liang et al., 2023; Huang et al., 2022; Gu et al., 2023) have been applied to this task. Unlike previous benchmarks based on *synthetic* environments, our benchmark uses real-world 3D

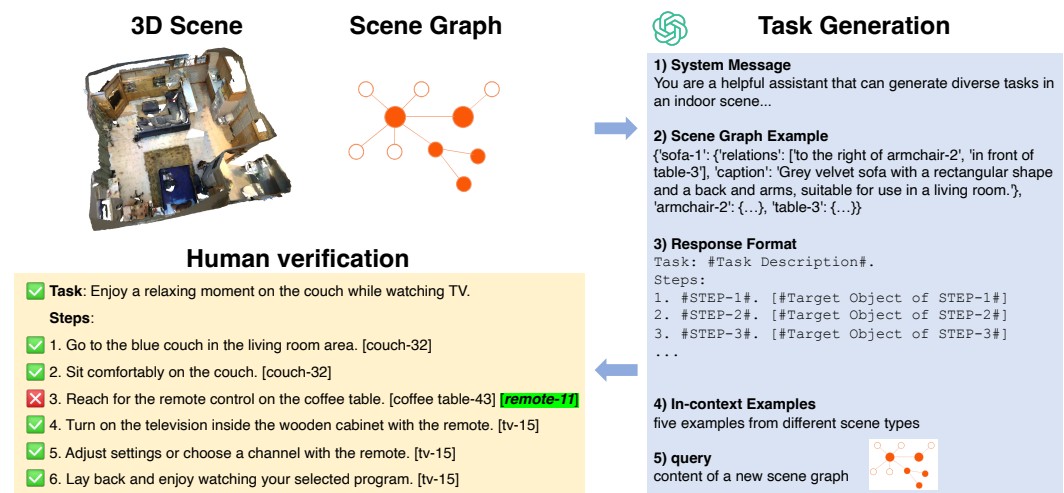

Figure 3: The pipeline of generating sequential grounding tasks in 3D scenes.

scenes, where noise, clutter, and missing or indistinguishable small objects in reconstructed point clouds make grounding more challenging than in cleaner, more controlled simulated environments. Moreover, by grounding each planned task to objects instead of low-level actions, we enable a broader range of actions and facilitate a more comprehensive analysis of results at each step.

**3D Large Language Model** Recent advancements in large language models (LLMs) have been significantly enhanced by integrating 3D spatial data, resulting in the development of 3D LLMs (Ma et al., 2024). Existing works, such as 3D-LLM (Hong et al., 2023) and Chat3D (Wang et al., 2023b), use object-centric or point-level representations to incorporate scene information into LLMs during instruction tuning (Hong et al., 2023; Xu et al., 2023; Li et al., 2024; Fu et al., 2024; Hong et al., 2024). LL3DA (Chen et al., 2023a) employs a Q-former-like (Li et al., 2023b) structure to further improve LLMs' 3D scene perception. Additionally, recent models like LEO (Huang et al., 2023a), 3D-VLA (Zhen et al., 2024), and ManipLLM (Li et al., 2023c) have introduced action capabilities into 3D LLMs, enabling them to interact with and manipulate objects in 3D environments (Huang et al., 2023a;b; Liu et al., 2024). Our work enhances the capabilities of 3D LLMs by incorporating grounding abilities, which output specific objects alongside the text.

## 3 THE 3D SEQUENTIAL GROUNDING BENCHMARK (SG3D)

### 3.1 PROBLEM FORMULATION

The problem of sequential grounding involves determining the relevance of objects in a given task. Specifically, given a 3D scene $\mathcal{S}$ and a task $\mathcal{T} = (t, \{a_1, ..., a_n\})$ where $t$ is a high-level task description and $a_1, ...a_n$ are detailed steps of the task plan, a model is required to predict a sequence of objects $\mathcal{O} = \{o_1, ..., o_n\}$, *i.e.*, the model needs to learn a mapping $f : (\mathcal{S}, \mathcal{T}) \rightarrow \mathcal{O}$. Compared to prior work, the challenge in our task lies in consistently grounding objects across sequential steps of a task plan.

### 3.2 DATASET CONSTRUCTION

As illustrated in Fig. 3, we leverage GPT-4 to generate tasks based on a 3D scene graph, followed by human verification. The full dataset is provided in the supplementary material.

**3D Scenes** Existing robotic task-planning approaches are typically evaluated in simulated environments (Shridhar et al., 2020; Li et al., 2023a; Rana et al., 2023), lacking observation of their effectiveness in real-world scenarios. To address this, we select reconstructed scenes as the 3D environment for our task. Specifically, we utilized real-world scenes from the SceneVerse dataset, incorporating scenes from ScanNet, ARKitScenes, HM3D, 3RScan, and MultiScan. In total, we

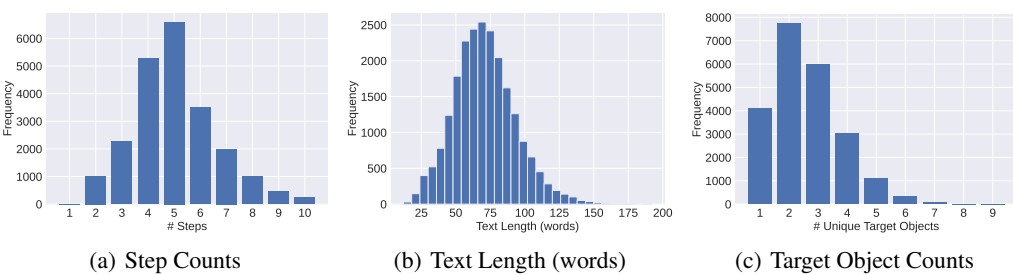

(a) Step Counts      (b) Text Length (words)      (c) Target Object Counts

Figure 4: Distributions of (a) step counts, (b) text length, and (c) target object counts per task.

curate 4,895 3D scenes in SG3D. Tab. 2 presents the number of scenes used in each dataset and the average number of object instances per scene.

**Scene Graphs** To provide GPT-4 with rich scene information, we process each scene into a semantic scene graph transformed from SceneVerse assets, which captures the categories, attributes, and spatial relations of objects within the scene. Each node in the graph represents a 3D object instance, while each edge represents a spatial relationship between nodes, such as "near", "below", or "embedded". We further enhance these scene graphs by adding object captions provided in SceneVerse.

**Task Generation** Using the 3D scene graph, we prompt GPT-4 (*gpt-4-turbo-2024-04-09*) to generate diverse tasks. For each scene, we ask GPT-4 to create five distinct tasks. Each task comprises a general description and several steps, with each step requiring the agent to focus on a specific target object, such as navigating toward or interacting with it. To ensure variety and coherence, we meticulously crafted the prompts and provided diverse in-context examples from multiple scene types. This approach guarantees that the generated tasks are both robust and varied. After generation, we remove any outputs with formatting errors and rigorously verify that all assigned targets are present in the corresponding scenes. Moreover, we observed that tasks exceeding ten steps tend to introduce hallucinated objects or problematic steps, which can negatively impact dataset quality. As a result, we discard any tasks containing more than ten steps. The detailed prompt used for GPT-4 can be found under Appendix A.1.

**Human Verification** We manually verify the evaluation set data to ensure quality. Given the 3D scene mesh and the task generated by GPT-4, annotators apply the following rules to judge each step's correctness:

1. If the step is unfeasible or unrelated to the task description, it is marked as incorrect.
2. If there is a missing step between step $k$ and step $k + 1$, step $k + 1$ is judged as incorrect.
3. When the step's description is insufficient to identify the target object, the step is considered correct if the target object can still be identified through context; otherwise, it is marked as incorrect.

Tasks with a single incorrect step are manually revised, while tasks containing more than one incorrect step are discarded. This human verification process ensures that the generated tasks are reasonable and the action steps are feasible. A screenshot of the interface for verification is provided under Appendix A.1.

### 3.3 DATASET ANALYSIS

In total, we collected data containing 22,346 tasks, encompassing 112,236 steps. Tab. 2 presents the statistics of task and step counts in our dataset. Each task description has an average length of 6.9 words, and each step has an average length of 12.7 words. The dataset was split into training and evaluation sets. For 3RScan, scenes from its training and evaluation splits were used as our training set, while scenes from its test split were used as the evaluation set. For other datasets, we adhered to the original split of the 3D scenes provided.

Fig. 4(a) illustrates the distribution of the number of steps per task, revealing an average of 5.03 steps per task. This underscores the complexity of our benchmark and the sequential nature of our data. Fig. 4(b) presents a histogram displaying the distribution of total text lengths for each task,

Table 2: Dataset statistics of SG3D.

| Dataset | #scenes | #obj. / scene | #tasks | #steps |
|---|---|---|---|---|
| 3RScan (Wald et al., 2019) | 472 | 31.5 | 2,194 | 11,318 |
| ScanNet (Dai et al., 2017) | 693 | 30.7 | 3,174 | 15,742 |
| MultiScan (Mao et al., 2022) | 117 | 40.8 | 547 | 2,683 |
| ARKitScenes (Baruch et al., 2021) | 1,575 | 12.1 | 7,395 | 39,887 |
| HM3D (Ramakrishnan et al., 2021) | 2,038 | 31.0 | 9,036 | 42,706 |
| Total | 4,895 | 25.1 | 22,346 | 112,336 |

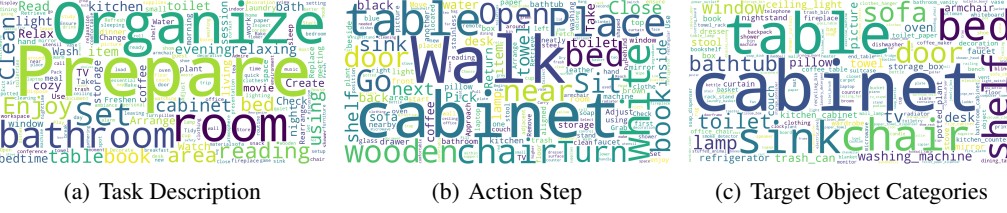

(a) Task Description      (b) Action Step      (c) Target Object Categories

Figure 5: Word clouds of (a) task description, (b) action step, and (c) target object categories.

including the task description and all steps, with an average of 70.5 words. This extended context poses a significant challenge for many text encoders, such as CLIP (Radford et al., 2021), indicating the need for models capable of handling lengthy inputs. Additionally, we examine the number of distinct target objects involved in each task, as shown in Fig. 4(c). Unlike the step counts, the number of unique target objects per task considers target objects with the same ID across different steps as one object, resulting in an average of 2.59 unique objects per task. This finding indicates that multiple objects are typically involved in this process.

To illustrate the diversity of our dataset, we present three word clouds here. Fig. 5(a) and Fig. 5(b) depict the frequency of words in task descriptions and action steps, respectively. In the task descriptions, the terms "prepare" and "organize" are the most prevalent activities. In the action steps, "walk" and "place" are the most common actions, "table" is the most frequent object, and "white" is the most frequent adjective. This indicates that task descriptions tend to be abstract and demand-oriented, while action steps offer detailed, execution-oriented instructions. Fig. 5(c) highlights the most frequently occurring target object categories, including but not limited to "cabinet", "table", "chair", "sink", "bed", "shelf", demonstrating the association of different object categories with the task guidance.

## 4 3D SEQUENTIAL GROUNDING MODELS

We explore several representative approaches for this purpose: three 3D-VL models depicted in Fig. 6—the dual-stream model 3D-VisTA (Zhu et al., 2023), the query-based model PQ3D (Zhu et al., 2024), the 3D LLM LEO (Huang et al., 2023a). Additionally, we investigate the integration of GPT-4 with an object labeler. Further details are provided in the subsequent discussion.

### 4.1 ARCHITECTURES

We follow ReferIt3D (Achlioptas et al., 2020) to use ground-truth object masks. To ensure a fair comparison, we employ the point cloud as the scene representation and the same PointNet++ (Qi et al., 2017) encoder to extract scene features for all three 3D-VL models.

**Dual-stream model.** In the dual stream model, we build upon the 3D-VisTA (Zhu et al., 2023) baseline. In 3D-VisTA, the model employs a spatial transformer to process 3D object representations and extracts text features using BERT (Devlin et al., 2018). These object and text tokens are then combined and fed into a unified transformer architecture to predict the target object. In our experiments, we concatenate the task description $t$ with detailed plans up to step $i$, $\{a_1, ..., a_i\}$, to serve as the textual input for predicting the target object $o_i$ at the current step.

Figure 6: Dual-stream model 3D-VisTA, query-based model PQ3D, and 3D LLM LEO.

**Query-based model.**     Unlike the dual stream model, the query-based model employs a generalized decoding framework for vision-language tasks (Zou et al., 2023; Zhu et al., 2020). PQ3D (Zhu et al., 2024) is a prominent query-based model designed for 3D environments, which unifies multiple representations and handles various tasks through multi-task training. This model leverages the CLIP (Radford et al., 2021) text encoder to process textual inputs. For a fair comparison with other models, we limit our implementation to the point feature branch for scene feature extraction. The input and output setting remain consistent with those of 3D-VisTA, as discussed above.

**3D LLM.**     The powerful reasoning capabilities of Large Language Models are highly advantageous for our task. We have adapted the recent 3D LLM LEO (Huang et al., 2023a) to suit our needs. In addition to predicting actions for each step, our model also predicts a special grounding token, [GRD]. This token is concatenated with object tokens and passed to the same grounding head used in 3D-VisTA and PQ3D to predict the target object, enabling integrated reasoning about both the previous and current step instructions. Unlike dual-stream and query-based models, which are constrained by their architectures and require separate forward passes for each action step, 3D LLM LEO concatenates $t$ and $\{a_1, ..., a_n\}$ to predict target objects for all steps sequentially by using multiple [GRD] tokens in a *single forward pass*.

**GPT-4 with an object labeler.**     In addition to the three 3D-VL models, we examine the applicability of GPT-4 for this task by integrating it with a PointNet++ (Qi et al., 2017) classifier, pre-trained on ScanNet, to predict semantic labels (categories) for objects. GPT-4 receives scene information in JSON format, which includes each object's ID, predicted category, center position, and size, along with the task description and detailed steps, tasked with generating a list of object IDs. Fig. A5 shows the specific prompt used here.

### 4.2    TRAINING & INFERENCE

During training, we optimize the three types of 3D-VL models using the cross-entropy loss, which compares the predicted object scores $f(\mathcal{S}, \mathcal{T})$ with the ground truth scores $\mathcal{O}$, as defined in Eq. (1). In the case of the 3D LLM, following the methodology of LEO, we introduce an additional cross-entropy loss to provide supervision for action generation in text format.

$$\mathcal{L}_{grd} = \mathbb{E}_{(\mathcal{S}, \mathcal{T}, \mathcal{O}) \sim \mathcal{D}} \text{CrossEntropy}(f(\mathcal{S}, \mathcal{T}), \mathcal{O}) \tag{1}$$

During inference, the 3D-VL models receive the task description and detailed steps, predicting the target object at each step. Beam search is utilized in LEO for generating action steps and the [GRD] token, with the beam width set to 5.

## 5    EXPERIMENTS AND RESULTS

### 5.1    SETTINGS

**Training Details**     We conduct training for all three 3D-VL model across all available datasets for 50 epochs. For optimization, we employ the AdamW optimizer, setting the learning rate at 1e-4, with $\beta_1$ configured to 0.9 and $\beta_2$ to 0.999. Additionally, we apply a weight decay of 0.05. Specifically, for the PQ3D and 3D-VisTA models, we utilize a batch size of 32. For the LEO model, we reduce the

batch size to 16 due to GPU memory limit. Furthermore, we use LoRA tuning (Hu et al., 2021) for the parameters of the LLM in LEO with a rank setting of 16.

**Evaluation Metrics** We assess the grounding performance of all models using two key metrics: task accuracy (t-acc) and step accuracy (s-acc). Task accuracy refers to the average grounding accuracy over the total number of tasks $t$. A sample is considered correct if the grounded objects are accurately identified for all steps within a task. Conversely, step accuracy is calculated by averaging the accuracy across all individual steps $a$. Task accuracy evaluates the model's ability to consistently interpret and respond accurately across a sequence of text prompts. On the other hand, step accuracy focuses on the model's effectiveness at each individual step.

## 5.2 QUANTITATIVE RESULTS & ANALYSIS

**1. Previous 3D Vision-Language models, such as dual-stream model 3D-VisTA and query-based model PQ3D, struggle to transfer to the sequential grounding task without fine-tuning.** As shown in Tab. 3, in the zero-shot setting, these models achieve low step accuracies ranging from 22.8% to 34.6% and task accuracies ranging from 0.0% to 10.3% across all datasets. This indicates that the models' pre-training on non-sequential tasks is insufficient for handling the complexities inherent in sequential grounding, highlighting the need for task-specific fine-tuning.

**2. Fine-tuning greatly enhances performance but low task accuracy scores (< 40%) indicate that consistent sequential grounding remains a challenge.** 3D-VisTA's t-acc increases from 8.3% to 30.6%, while PQ3D's t-acc improves from 7.8% to 26.8%. The 3D LLM model LEO achieves the best performance after fine-tuning, with a s-acc of 62.8% and a t-acc of 34.1%. Despite these improvements, all models' t-acc scores remain below 40%, indicating that current models still struggle to achieve consistent sequential grounding. This limitation highlights the need for further research and model design to effectively address the challenges posed by sequential grounding tasks.

**3. The 3D LLM model, LEO, consistently outperforms the other models across all datasets, particularly in terms of task accuracy.** LEO achieves the highest task accuracies 34.1%, compared to 3D-VisTA 30.6% and PQ3D 26.8%. This advantage can be attributed to LEO's 3D LLM architecture, which effectively captures and reasons about sequential dependencies in grounding tasks. Although LEO also enhances step accuracy, the improvement is less substantial compared to the significant gains observed in task accuracy.

**4. The combination of GPT-4 and 3D object classifier is insufficient for addressing the sequential grounding task.** Despite GPT-4's robust reasoning and generalization capabilities, its performance—recording a t-acc of 7.6% and a s-acc of 27.3%—is significantly inferior to that of fine-tuned 3D vision-language models. This shortfall can be attributed to classification inaccuracies and the loss of information when translating the scene into semantic labels and positions. These results indicate that the effectiveness of large language models in this problem is heavily influenced by the alignment between 3D vision modality and text modality, making 3D-VL models the more suitable approach.

## 5.3 ABLATION STUDY

**Effect of offering contextual information.** To investigate the impact of contextual information, we eliminate multi-step action context during both the training and inference phases, providing only the task description and current action step. The experimental results illustrated in Fig. 7 reveal a significant decline in task accuracy upon the removal of contextual information for both LEO and 3D-VisTA. Specifically, LEO exhibits an average t-acc drop of 3.4%, while 3D-VisTA demonstrates an even more pronounced decline of 5.0%. This suggests that the models have, to a certain extent, learned to leverage contextual information during the grounding process. In contrast, PQ3D experiences a more modest performance reduction, with an average t-acc decrease of only 0.8%. This limited decline can be attributed to PQ3D's reliance on a CLIP text encoder, which struggles to interpret lengthy sentences (Zhang et al., 2024), thereby leading to overfitting on shorter, single-step instructions.

**Impact of training data volume and data efficiency comparison.** Fig. 8 shows that increasing the volume of training data utilized during fine-tuning improves the performance of all three models. Notably, LEO exhibits superior data efficiency, achieving comparable performance to PQ3D and

Table 3: **The grounding accuracy on SG3D.** "s-acc" denotes the grounding accuracy averaged over steps and "t-acc" denotes the grounding accuracy averaged over tasks. A task is considered correct if and only if all steps are correct. We run each experiment three times and report error bars.

| | Model Type | ScanNet | | 3RScan | | MultiScan | |
|---|---|---|---|---|---|---|---|
| | | s-acc (%) | t-acc (%) | s-acc (%) | t-acc (%) | s-acc (%) | t-acc (%) |
| **Zero-shot** | | | | | | | |
| 3D-VisTA | *Dual-stream* | 26.9 | 4.7 | 23.7 | 2.2 | 22.8 | 4.7 |
| PQ3D | *Query-based* | 29.7 | 4.1 | 24.6 | 2.9 | 23.2 | 0.0 |
| GPT-4 w/ pred labels | *LLM* | 42.6 | 10.9 | 25.5 | 2.4 | 27.0 | 0.0 |
| **Fine-tune** | | | | | | | |
| 3D-VisTA | *Dual-stream* | $58.4 \pm 0.1$ | $21.1 \pm 0.5$ | $53.3 \pm 0.8$ | $14.9 \pm 1.5$ | $48.3 \pm 3.4$ | $\mathbf{11.6 \pm 2.4}$ |
| PQ3D | *Query-based* | $54.8 \pm 0.8$ | $17.8 \pm 0.7$ | $49.3 \pm 1.3$ | $9.9 \pm 2.5$ | $46.4 \pm 2.1$ | $4.7 \pm 0$ |
| LEO | *3D LLM* | $\mathbf{61.2 \pm 1.0}$ | $\mathbf{25.7 \pm 1.7}$ | $\mathbf{55.8 \pm 0.6}$ | $\mathbf{16.0 \pm 1.8}$ | $\mathbf{52.7 \pm 1.6}$ | $7.6 \pm 1$ |

| | Model Type | ARKitScenes | | HM3D | | OverAll | |
|---|---|---|---|---|---|---|---|
| | | s-acc (%) | t-acc (%) | s-acc (%) | t-acc (%) | s-acc (%) | t-acc (%) |
| **Zero-shot** | | | | | | | |
| 3D-VisTA | *Dual-stream* | 30.8 | 9.0 | 25.3 | 10.3 | 26.9 | 8.3 |
| PQ3D | *Query-based* | 34.6 | 8.6 | 24.4 | 9.7 | 28.2 | 7.8 |
| GPT-4 w/ pred labels | *LLM* | 27.6 | 6.0 | 20.8 | 7.7 | 27.3 | 7.6 |
| **Fine-tune** | | | | | | | |
| 3D-VisTA | *Dual-stream* | $68.8 \pm 0.9$ | $37.6 \pm 1.1$ | $59.6 \pm 0.7$ | $32.4 \pm 0.8$ | $60.9 \pm 0.4$ | $30.6 \pm 0.7$ |
| PQ3D | *Query-based* | $65.2 \pm 0.5$ | $32.1 \pm 0.7$ | $56.1 \pm 0.3$ | $30.0 \pm 0.7$ | $57.3 \pm 0.1$ | $26.8 \pm 0.5$ |
| LEO | *3D LLM* | $\mathbf{69.6 \pm 0.4}$ | $\mathbf{41.5 \pm 1.5}$ | $\mathbf{61.5 \pm 1}$ | $\mathbf{35.7 \pm 1.3}$ | $\mathbf{62.8 \pm 0.7}$ | $\mathbf{34.1 \pm 1.2}$ |

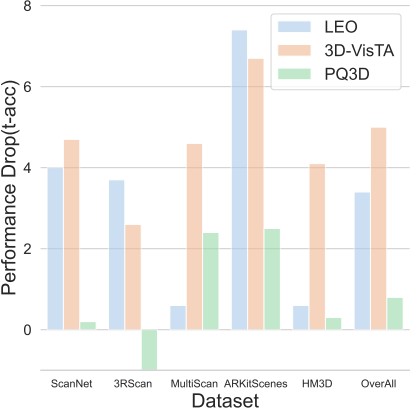

Figure 7: Ablation of contextual information.

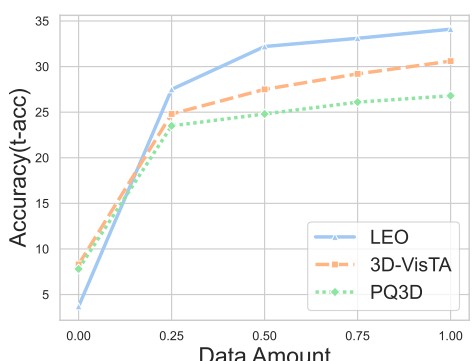

Figure 8: Impact of training data volume and data efficiency comparison.

3D-VisTA using only 25% of the data. This advantage is likely attributable to LEO's foundation on a large language model, which has been pre-trained on a vast array of task-relevant information and acquired common-sense knowledge.

## 5.4 QUALITATIVE RESULTS

Fig. 9 demonstrates that sequential grounding tasks require models to reason across sequential steps. The results from LEO show that after training, the model is capable of performing sequential grounding, as evidenced in tasks 1, 2, and 5. However, the model sometimes struggles to maintain sequential consistency across sequential steps, as observed in task 3. Task 4, in particular, highlights a failure case in which the model fails to grasp the concept of a diaper bin. The examples highlight the challenges and complexities inherent in sequential grounding tasks, emphasizing the need for models possessing both robust sequential reasoning abilities and a solid understanding of common sense knowledge to achieve consistent and accurate results.

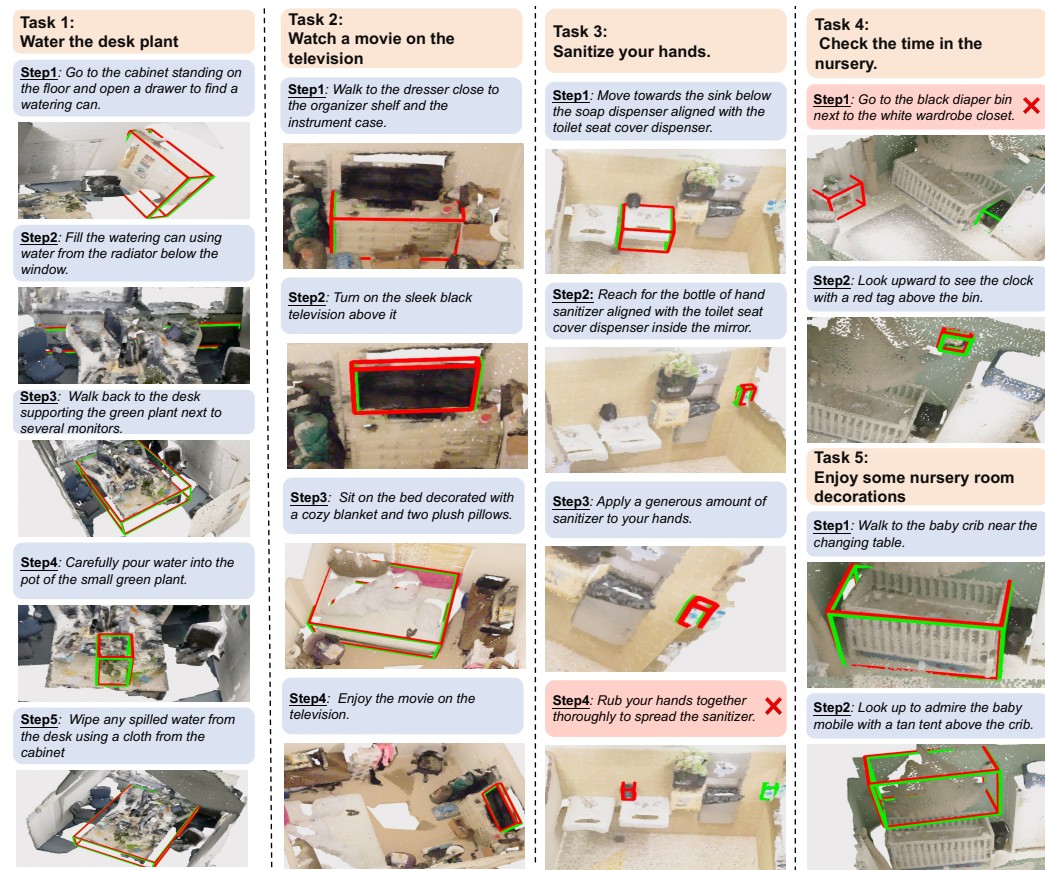

Figure 9: **Qualitative results from LEO.** Red are predictions and green are ground-truth boxes.

## 6 CONCLUSION

In this work, we introduce the task of Task-oriented Sequential Grounding in 3D scenes and present SG3D, a large-scale dataset designed to facilitate research in this area. Evaluations of state-of-the-art 3D visual grounding models on SG3D benchmark reveal the substantial challenges in adapting these models to sequential grounding tasks. These results emphasize the necessity for further research and model development. We encourage the community to move beyond traditional 3D visual grounding towards more practical, task-oriented applications, paving the way for more advanced and capable embodied agents.

## 7 DISCUSSIONS

**Rationale for limiting to one target object per step**    The primary consideration for this restriction is that most mobile manipulators (e.g., the one used in SayCan (Ahn et al., 2022)) are *single-arm* and can manipulate only one object at a time. This design aligns with current practical constraints and facilitates the adaptation of 3D visual grounding models to real-world robotic tasks. Nevertheless, our data generation pipeline is flexible and can be easily adapted to support multi-target actions by adjusting the GPT-4 prompt, as detailed in Fig. A1.

**Handling steps that do not appear to involve a target object**    Some steps, like "Rub your hands" (task 3's step 4 in Fig. 9), involving the agent itself rather than a specific object in the scene, we consider the target object from the previous step as the reference. This implies that no positional change is required, which is a reasonable assumption in the navigation setting. These steps reflect realistic interactions and are part of the task's natural sequence, so we keep them in our dataset.

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

# A APPENDIX

## A.1 DETAILS OF DATASET CONSTRUCTION

**Detailed Prompt used in Task Generation**   The prompt messages employed in the task generation process are depicted in Fig. A1, with the "System prompt" specifically illustrated in Fig. A2. Specific in-context examples, denoted as "<EXAMPLES>" in the system prompt, are presented in Fig. A3. We deliberately omit to show GPT-4 the corresponding scene graph for the provided response examples, as an overly long context increases the likelihood of errors.

> messages = [{'role': 'system', 'content': System prompt}, {'role': 'user', 'content': Scene graph of the scene to process}]

Figure A1: Prompts messages for GPT-4 task generation.

**Details in Human Verification**   Fig. A4 shows the interface used for human verification. The interface consists primarily of an interactive 3D mesh window and a right-hand column that displays task data. When a specific step is selected, the target object is highlighted within the mesh using a red bounding box. Users can rotate, translate, and zoom in or out within the 3D mesh window. Annotators mark each step with a tick or a cross. Following this verification process, tasks containing one incorrect step are manually revised.

## A.2 ADDITIONAL DATA STATISTICS

The statistics for task and step counts in the training and evaluation splits are presented separately in Tab. A1.

Table A1: Statistics of the training and evaluation splits for various datasets.

|  |  | Training Set | Evaluation Set | Train+Eval |
|---|---|---|---|---|
| 3RScan | # tasks | 2,056 | 138 | 2,194 |
|  | # steps | 10,622 | 696 | 11,318 |
| ScanNet | # tasks | 2,731 | 443 | 3,174 |
|  | # steps | 13,634 | 2,108 | 15,742 |
| MultiScan | # tasks | 504 | 43 | 547 |
|  | # steps | 2,459 | 224 | 2,683 |
| ARKitScenes | # tasks | 6,952 | 443 | 7,395 |
|  | # steps | 37,552 | 2,335 | 39,887 |
| HM3D | # tasks | 8,146 | 890 | 9,036 |
|  | # steps | 38,833 | 3,873 | 42,706 |
| Total | # tasks | 20,389 | 1,957 | 22,346 |
|  | # steps | 103,100 | 9,236 | 112,336 |

## A.3 ADDITIONAL INFORMATION OF BASELINES

**Detailed Prompt used in the GPT-4 baseline**   Fig. A5 details the prompt messages employed in the baseline of GPT-4 integrated with an object labeler.

**Benchmarking more baselines**   We also evaluated more grounding baselines, including MiKASA-3DVG (Chang et al., 2024), ViewRefer (Guo et al., 2023), and Vil3DRef (Chen et al., 2022a), on SG3D to validate the robustness of our findings. To maintain consistency with our main experiments, we trained all these models for 50 epochs. For ease of comparison, we include these results alongside

> You are a helpful assistant that can generate diverse tasks in an indoor scene.
>
> The scene is represented by a scene graph in the JSON dictionary format. Each entity in the scene graph denotes an object instance, named '<category>-<ID>'. The 'caption' describes the object's attributes, such as 'color', 'material', etc. The 'relations' describes the object's spatial relations with other objects. For example, from the scene graph:
> ```
> 'sofa-1': 'relations': ['to the right of armchair-2', 'in front of table-3'], 'caption': 'Grey velvet sofa with a rectangular shape and a back and arms, suitable for use in a living room.', 'armchair-2': 'relations': ['to the left of sofa-1'], 'caption': 'The armchair is made of leather, specifically black leather, and has a spherical shape.', 'table-3': 'relations': [], 'caption': 'The table is a rectangular wooden table with a brown finish, sometimes used as a dining table or coffee table, with a smooth wooden texture and various styles, including a sign or place setting on it, and can have plates or a white cloth on it.'
> ```
>
> You can know that 'sofa-1' is grey, the 'armchair-2' is made of leather, the 'table-3' is made of wood, the 'armchair-2' is on the left of the 'sofa-1', the 'sofa-1' is in front of the 'table-3'.
>
> Using the provided scene graph, design daily tasks that a person can do in this scene. Besides, decomposing every task into a sequence of steps that can be performed using the objects in this scene. For each step, give the target object that the person should attend to. Your output must follow the template below:
> ```
> Task: #Describe the task using one sentence.#
> Steps:
> 1. #The step must perform only one action. Split actions such as 'pick up xxx and place it xxx' into two separate steps. All objects, attributes, and relations must be explicitly listed in the given scene graph. Do not include the IDs of the objects, use ordinal words, attributes, and relations to refer to different object instances of the same category. Use pronouns ('it', 'them', 'here', and 'the other', etc.) as much as possible to make the step concise.# [#Use '<category>-<ID>' to denote the target object. Do NOT assume objects that do not exist in the scene graph! Each step must have exactly one target object. #]
> 2. ...
> 3. ...
> ...
> ```
>
> Here are some examples:
> ```
> <EXAMPLES>
> ```
>
> Generate 5 different tasks involving different objects and separate these tasks by "===".

Figure A2: System prompt for GPT-4 task generation.

our main experimental results in Tab. A2. The results further support the conclusions we proposed in the manuscript.

## A.4 HUMAN STUDY

We conducted a human study by randomly selecting 100 tasks from the evaluation set. Participants were given an interactive 3D scene and a task in a web viewer. Despite some artifacts in the 3D scene viewer, human participants achieved 85% step accuracy and 63% task accuracy, significantly outperforming baseline models. This demonstrates that the proposed task and dataset are indeed challenging for current models. A series of screenshots of the human study interface to demonstrate a human study case is provided in the supplementary material for reference.

Table A2: **The grounding accuracies of more baselines on SG3D.** "s-acc" denotes the grounding accuracy averaged over steps and "t-acc" denotes the grounding accuracy averaged over tasks. A task is considered correct if and only if all steps are correct.

| | ScanNet | | 3RScan | | MultiScan | |
| --- | --- | --- | --- | --- | --- | --- |
| | s-acc (%) | t-acc (%) | s-acc (%) | t-acc (%) | s-acc (%) | t-acc (%) |
| 3D-VisTA | 58.4 | 21.1 | 53.3 | 14.9 | 48.3 | **11.6** |
| PQ3D | 54.8 | 17.8 | 49.3 | 9.9 | 46.4 | 4.7 |
| LEO | **61.2** | **25.7** | **55.8** | **16.0** | 52.7 | 7.6 |
| MiKASA-3DVG | 57.8 | 20.3 | 53.0 | 10.9 | 48.7 | 2.3 |
| ViewRefer | 59.9 | 20.8 | 54.6 | 6.5 | 48.7 | 4.7 |
| Vil3DRef | 60.2 | 20.8 | 53.3 | 11.6 | **53.6** | **11.6** |

| | ARKitScenes | | HM3D | | OverAll | |
| --- | --- | --- | --- | --- | --- | --- |
| | s-acc (%) | t-acc (%) | s-acc (%) | t-acc (%) | s-acc (%) | t-acc (%) |
| 3D-VisTA | 68.8 | 37.6 | 59.6 | 32.4 | 60.9 | 30.6 |
| PQ3D | 65.2 | 32.1 | 56.1 | 30.0 | 57.3 | 26.8 |
| LEO | 69.6 | **41.5** | **61.5** | **35.7** | **62.8** | **34.1** |
| MiKASA-3DVG | 66.4 | 33.6 | 57.2 | 30.6 | 59.1 | 26.9 |
| ViewRefer | 68.2 | 34.8 | 57.3 | 30.0 | 60.2 | 27.9 |
| Vil3DRef | **70.1** | 37.5 | 58.0 | 29.7 | 61.0 | 29.0 |

## A.5 IMPACT ON EMBODIED NAVIGATION

While interactive evaluation of action sequences is currently infeasible due to the static nature of the reconstructed 3D scenes, we demonstrate the relevance of our annotations by integrating the LEO model with a navigation module in an embodied setting. Specifically, we use the GreedyGeodesicFollower class from Habitat-Sim (Savva et al., 2019) to guide task-oriented navigation within HM3D scenes based on the grounding results (the centers of the target objects). We have provided three navigation videos showcasing this process in the supplementary material.

## A.6 PLANNING ABILITY OF LEO

In this additional experiment, we evaluate the planning ability of LEO fine-tuned on SG3D. Given a task description $t$, LEO is required to generate both action steps $\{a_1, .., a_n\}$ and target objects $\{o_1, ..., o_n\}$. Since action steps can be rearranged into various topological orders, we do not employ exact matches to assess the similarity between the predicted and ground truth plans. Instead, we utilize metrics from OpenEQA (Majumdar et al., 2024), which leverage GPT-4 to score the model's output based on ground truth. A score of 1 indicates no match, while a score of 5 represents a perfect match. In our experiments, the GPT score on ScanNet is $2.1 \pm 1.0$, suggesting considerable room for improvement. The prompts used for score computation are detailed in Fig. A6.

Task: Make me a cup of coffee.
Steps:
1. Go to the long desk against the wall. [desk-15]
2. Choose a cup from those white, plastic cups on the top of the desk. [cups-19]
3. Fill it with coffee at the coffee maker. [coffee maker-16]
4. Walk to the table close to a cabinet. [table-23]
5. Put the cup down. [table-23]
6. Return to the long desk. [desk-15]
7. Fetch a plate from a bunch of steel plates below a picture frame hanging on the wall. [plates-17]
8. Go back to the table. [table-23]
9. Put the cup on the plate on the table. [table-23]
===
Task: Watch tv from the sofa.
Steps:
1. Go to the black table to the left of the fire extinguisher. [table-30]
2. Grab the black remote lying on it. [remote-36]
3. Turn on the tv with the remote. [tv-38]
4. Walk to the table in the middle of the bed and the white cabinet. [table-58]
5. Place the remote here. [table-58]
6. Walk to the black sofa close to the bed. [sofa-14]
7. Sit here to admire tv show. [sofa-14]
===
Task: Clean the mirror.
Steps:
1. Walk to the white cabinet. [cabinet-7]
2. Grab the towel on it. [towel-10]
3. Put the towel into the sink. [sink-37]
4. Turn the faucet on. [faucet-13]
5. Wet the towel in the sink. [sink-37]
6. Turn the faucet off. [faucet-13]
7. Wipe the mirror with the towel. [mirror-11]
8. Put the towel into the sink again. [sink-37]
9. Turn the faucet on. [faucet-13]
10. Wash the towel in the sink. [sink-37]
11. Turn the faucet off. [faucet-13]
12. Wring the towel dry in the sink. [sink-37]
13. Put it back to the cabinet. [cabinet-7]
===
Task: Browse the internet.
Steps:
1. Walk to the desk adorned with papers. [desk-19]
2. Turn on the computer tower behind the desk and the bookshelf. [computer tower-7]
3. Sit down on the nearest chair. [chair-26]
4. Fetch the mouse on the desk. [mouse-8]
5. Look at the screen of the monitor. [monitor-14]
===
Task: Go to sleep.
Steps:
1. Go to the curtain. [curtain-11]
2. Close it. [curtain-11]
3. Walk to the nightstand with the telephone. [nightstand-15]
4. Turn off the lamp on this nightstand. [lamp-19]
5. Go to the other nightstand. [nightstand-14]
6. Set the alarm on it. [alarm clock-28]
7. Lie down on the bed. [bed-20]

Figure A3: <EXAMPLES> in system prompt for GPT-4 task generation.

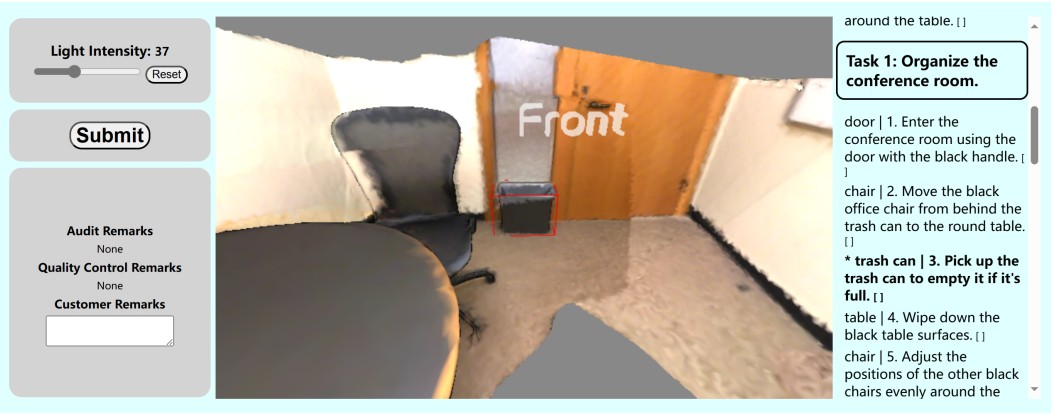

Figure A4: Screenshot of the interface for human verification.

```
# system prompt (role: system)
You are tasked with identifying the target object for each step in a given task. Each scene contains various objects,
and your response should provide the target object for each step in the format <label-id>, maintaining the sequence
of steps. For example:

# example task (role: user)
Task: Make me a cup of coffee and serve it on a plate.
Steps:
1. Go to the long desk against the wall.
2. Fetch a plate from a bunch of steel plates below the picture frame.
3. Walk to the table close to a cabinet.
4. Put the plate on it.
5. Return to the long desk.
6. Choose a cup from those white, plastic cups on the desk.
7. Fill it with coffee at the coffee maker.
8. Go back to the table.
9. Put down the cup of coffee.

# example scene (role: user)

"table-24":
"position": [
-4.913224259334377,
2.2510899724225615,
-0.9699999988079071
],
"size": [
2.032371906039741,
1.247916508679886,
0.8399999737739563
]
,
...

# example response (role: assistant) 1. desk-15
2. plates-17
3. table-23
4. table-23
5. desk-15
6. cups-19
7. coffee maker-16
8. table-23
9. table-23

# role: user
< CURRENT TASK & SCENE >
```

Figure A5: Prompt messages used in the GPT-4 baseline.

You are a helpful assistant that can evaluate the quality of task planning given a scene, a task description, a ground truth task planning, and a predicted task planning. To mark a response, you should output a single integer between 1 and 5 (including 1, 5), with format ```Your mark: number```. 5 means that the predicted task planning perfectly solves the problem described in the task and matches the ground truth task planning. 1 means that the predicted task planning is completely irrelevant to the task description and does not match the ground truth task planning.

The scene is represented by a scene graph in the JSON dictionary format. Each entity in the scene graph denotes an object instance, named '<category>-<ID>'. The 'caption' describes the object's attributes, such as 'color', 'material', etc. The 'relations' describes the object's spatial relations with other objects. For example, from the scene graph:
```
'sofa-1': 'relations': ['to the right of armchair-2', 'in front of table-3'], 'caption': 'Grey velvet sofa with a rectangular shape and a back and arms, suitable for use in a living room.', 'armchair-2': 'relations': ['to the left of sofa-1'], 'caption': 'The armchair is made of leather, specifically black leather, and has a spherical shape.', 'table-3': 'relations': [], 'caption': 'The table is a rectangular wooden table with a brown finish, sometimes used as a dining table or coffee table, with a smooth wooden texture and various styles, including a sign or place setting on it, and can have plates or a white cloth on it.'
```

You can know that 'sofa-1' is grey, the 'armchair-2' is made of leather, the 'table-3' is made of wood, the 'armchair-2' is on the left of the 'sofa-1', the 'sofa-1' is in front of the 'table-3'.

Using the provided scene graph, you should decide whether predicted task planning can solve the problem described in task description.
Here are some examples:
```
<example>
```

Your Turn, output with format ```Your mark: number```.
Scene graph: <scene graph>
Task description: <task description>
Ground truth task planning text: <gt plan text>
Ground truth object id: <gt object id>
Predicted task planning text: <pred plan text>

Figure A6: Prompt messages for computing GPT score.

