# OpenReview forum: "Task-oriented Sequential Grounding in 3D Scenes"
_ICLR.cc/2025/Conference — Submitted to ICLR 2025_

### Official Review · Reviewer_RYps · 2024-10-30

**Soundness:** 2
**Presentation:** 2
**Contribution:** 2
**Rating:** 5
**Confidence:** 4

**Summary:**

This paper presents a novel benchmark, SG3D (Task-oriented Sequential Grounding in 3D scenes), designed for task-oriented sequential grounding in 3D environments. This dataset, built on 3D scene data from sources such as ScanNet, ARKitScenes, and HM3D, includes 22,346 tasks with over 112,000 steps across 4,895 3D scenes. Unlike traditional 3D grounding tasks, SG3D emphasizes the sequential, task-oriented nature of grounding, where an agent must follow detailed multi-step instructions, each step requiring context-based object identification. The authors adapted existing models, including 3D-VisTA, PQ3D, and LEO, to this new setting, demonstrating that these models face significant challenges under SG3D’s task requirements. Their work highlights the gap in current models' ability to perform consistent, sequential grounding in complex environments, underscoring the need for specialized approaches.

**Strengths:**

The SG3D dataset is built on a diverse foundation, incorporating 4,895 3D scenes spanning five distinct categories, resulting in a robust and varied dataset. The task-centered design of textual instructions, focused on guiding sequential actions rather than mere object identification, is particularly valuable.

**Weaknesses:**

The experimental evaluation is somewhat unclear, making it difficult to discern whether the model outputs bbx or target IDs. If the output is bounding boxes, the supplementary material lacks the corresponding label data, while if it is target IDs, the origin of the bounding boxes in Figure 9 remains unexplained.
The proposed method outputs only target-specific information at each step, without addressing movement between steps, which raises questions about continuity. Specifically, is the scene input 𝑆 the same for each step? If so, this would contradict the perspective shifts caused by position changes in real-world applications.
Additionally, the dataset's purpose seems limited and somewhat misaligned. Fundamentally, it focuses on grounding tasks, where multiple steps appear unnecessary and potentially confusing, with their only purpose seeming to ensure consistency across objects. Without a memory mechanism, ensuring consistency without knowledge of prior steps feels unconvincing. The dataset thus lacks a strong rationale for multi-step grounding, as it does not effectively integrate planning or contextual continuity.

**Questions:**

Is the scene input 𝑆 the same for each single-step grounding?

---

> ### Author Response · Authors · 2024-11-21
> **Response to Reviewer RYps**
>
> We would like to thank the reviewer for the thoughtful evaluation of our manuscript. We appreciate the constructive feedback provided and will address each of the mentioned limitations point by point.
>
> > **W1:** The experimental evaluation is somewhat unclear, making it difficult to discern whether the model outputs bbx or target IDs. If the output is bounding boxes, the supplementary material lacks the corresponding label data, while if it is target IDs, the origin of the bounding boxes in Figure 9 remains unexplained.
>
> The model outputs are target IDs, since we use ground-truth object masks, as clarified in Line 315. The bounding boxes shown in Figure 9 are calculated from these ground-truth object masks and are included solely for result visualization purposes.
>
> > **W2.1:** The proposed method outputs only target-specific information at each step, without addressing movement between steps, which raises questions about continuity.
>
> Our work focuses on the visual grounding task, aiming to localize the target object described by a sentence within a scene. Consequently, the method outputs target-specific information at each step without addressing movement between steps, as this pertains more to navigation tasks. While handling movement is indeed important, it lies beyond the scope of this study. We acknowledge its significance and plan to explore this aspect in future research.
>
> > **W2.2:** Specifically, is the scene input 𝑆 the same for each step? If so, this would contradict the perspective shifts caused by position changes in real-world applications.
>
> Yes, the scene input 𝑆 is the same for each step. This is consistent with the visual grounding task, which relies on a global scene input to localize target objects. Unlike navigation tasks, which account for perspective shifts using ego-view images from a robot's perspective, our setting involves a static scene without real robots. Thus, perspective shifts are not applicable in our approach.
>
> > **W3:** Additionally, the dataset's purpose seems limited and somewhat misaligned. Fundamentally, it focuses on grounding tasks, where multiple steps appear unnecessary and potentially confusing, with their only purpose seeming to ensure consistency across objects. Without a memory mechanism, ensuring consistency without knowledge of prior steps feels unconvincing. The dataset thus lacks a strong rationale for multi-step grounding, as it does not effectively integrate planning or contextual continuity.
>
> While dual-stream models like 3D-VisTA and query-based models like PQ3D lack memory mechanisms, the 3D LLM LEO incorporates such a mechanism. Specifically, LEO utilizes a grounding token [GRD] to represent each step's prediction, as demonstrated in Figure 6. Because of its autoregressive structure, LEO’s predictions for previous steps directly influence predictions for subsequent steps, thereby enabling consistency across steps and providing a rationale for multi-step grounding.
>
> > **Q1:** Is the scene input 𝑆 the same for each single-step grounding?
>
> Yes, the scene input 𝑆 remains the same for each single-step grounding, as explained in our response to **W2.2**.

---

> > ### Comment · Reviewer_RYps · 2024-11-22
> >
> > If the output is the predicted ID, the bounding box should correspond to the ground-truth bounding box of the predicted ID. How, then, are the subtle differences between the red and green bounding boxes in Figure 9 generated?

---

> > > ### Author Response · Authors · 2024-11-26
> > > **Clarification on bounding box visualization**
> > >
> > > > **Q2:** If the output is the predicted ID, the bounding box should correspond to the ground-truth bounding box of the predicted ID. How, then, are the subtle differences between the red and green bounding boxes in Figure 9 generated?
> > >
> > > When the predicted ID matches the ground-truth ID, the bounding boxes are identical, as they are both derived from the ground-truth object mask. The subtle differences between the red and green bounding boxes in Figure 9 are simply for visualization purposes, as two boxes overlap.
> > >
> > > We appreciate the reviewer's request for clarification and will make this explanation clearer in the revised paper.

---

> > > > ### Comment · Reviewer_RYps · 2024-11-27
> > > >
> > > > I thank the author for their reply. After reading the rebuttal, I'm keeping my score.

---

> ### Comment · Reviewer_RYps · 2024-11-22
>
> Although line 74 emphasizes that **"To solve this task, an agent must understand each step in the context of the whole plan to identify the target object, since a single step alone can be insufficient to distinguish the target from other objects of the same category,"** the dataset construction and experiment seem to lack sufficient emphasis on leveraging contextual information. This undermines the overall persuasiveness of the approach.

---

> > ### Author Response · Authors · 2024-11-27
> > **Clarification on leveraging contextual information**
> >
> > > **Q3:** Although line 74 emphasizes that "To solve this task, an agent must understand each step in the context of the whole plan to identify the target object, since a single step alone can be insufficient to distinguish the target from other objects of the same category," the dataset construction and experiment seem to lack sufficient emphasis on leveraging contextual information. This undermines the overall persuasiveness of the approach.
> >
> > We sincerely thank the reviewer for highlighting the importance of leveraging contextual information in our approach. Reviewer CNtV raised a similar concern, which we have addressed by providing *qualitative examples* from the dataset and *quantitative results* from our ablation studies (see our responses to Reviewer CNtV's **Q1.1**, **Q1.2**, and **Q1.3**). These additions clarify how contextual information is utilized and should help address your concern as well. Additionally, we will refine our manuscript to better emphasize how contextual information is utilized, incorporating this valuable suggestion.

---

> > > ### Comment · Reviewer_RYps · 2024-11-27
> > >
> > > In your response to Reviewer CNtV's Q1.2, you mentioned that all three tested models performed better when context was provided. However, as you stated, 3D-VisTA and PQ3D do not have mechanisms to process context. This raises concerns about the reliability of the observed improvement. Additionally, the input for these two models does not seem to contain contextual information. Could you clarify where this so-called "context" originates during the models' inference process?

---

> > > > ### Author Response · Authors · 2024-11-27
> > > > **Clarification on context**
> > > >
> > > > > **Q4:** In your response to Reviewer CNtV's Q1.2, you mentioned that all three tested models performed better when context was provided. However, as you stated, 3D-VisTA and PQ3D do not have mechanisms to process context. This raises concerns about the reliability of the observed improvement. Additionally, the input for these two models does not seem to contain contextual information. Could you clarify where this so-called "context" originates during the models' inference process?
> > > >
> > > > **Context** refers to information from the task description and prior step instructions. In our experiments, we provided all models with this context by including both the task description and instructions for all prior steps at each step's grounding, rather than just the current step.
> > > >
> > > > While we stated that 3D-VisTA and PQ3D lack memory mechanisms, this does not mean they cannot process context. For these models, *each action step requires an independent forward pass*. At every step, the text input includes the task description, prior step instructions, and the current step's instruction. However, these models cannot track or use their previous decisions across steps, which limits their ability to achieve consistent grounding.
> > > >
> > > > In contrast, *3D LLM LEO predicts target objects for all steps sequentially in a single forward pass*. As illustrated in Figure 6, it employs grounding tokens ([GRD]s) to represent its choices at each step, enabling it to retain and leverage decisions from prior steps.
> > > >
> > > > We hope this clarification addresses your concern.

---

> > > > > ### Comment · Reviewer_RYps · 2024-11-28
> > > > >
> > > > > Alright, I have no further questions. I will adjust my score back to my initial score of 5 to encourage the authors to continue their exploration. However, to be honest, my personal inclination leans towards a score of 4. Best of luck!

---

> > > > > > ### Author Response · Authors · 2024-11-28
> > > > > > **Acknowledgement**
> > > > > >
> > > > > > Thank you for your thoughtful review and for taking the time to carefully consider our rebuttal. We sincerely appreciate your encouragement and feel inspired to explore this task further and refine our manuscript.

---

> ### Comment · Reviewer_RYps · 2024-11-22
>
> I believe that the motivation stated in the paper does not align cohesively with the proposed dataset and method, failing to form a convincing logical loop. As a result, I will lower the score to 3.

---

### Official Review · Reviewer_3oG2 · 2024-11-01

**Soundness:** 3
**Presentation:** 4
**Contribution:** 3
**Rating:** 6
**Confidence:** 3

**Summary:**

This paper introduces Task-oriented Sequential Grounding, where a long-horizon task with detailed steps is given, and the objects of interest need to be localized considering not only the current action step but the entire task context. This paper also introduces SG3D, a large-scale dataset for the above task, which includes over 22k tasks and 112k steps. This paper also finetunes and evaluates 3 typical 3D-VL models on SG3D, including a dual-stream model, a query-based model, and a 3D LLM, as well as the GPT-4 with an object labeler. Results show that these models struggle to transfer to SG3D without finetuning. After finetuning, their performance is still limited (<40%). This shows the open challenges existing in the proposed task and dataset.

**Strengths:**

1. The paper proposes the sequential grounding task and a new dataset for this task. Compared to existing visual grounding tasks, sequential grounding needs the agent to incorporate more environmental and instructional context. This is more challenging in terms of 3D visual grounding.

2. The paper provides a detailed analysis of the introduced dataset.

3. This paper evaluates several existing 3D vision-language model baselines on the proposed task. The results show that these models only show limited performance, and there is a research gap to be filled. Ablation studies also show the importance of contextual information and the effect of the amount of training data.

4. This paper is well-written and easy to follow.

**Weaknesses:**

1. The definition of the sequential grounding task is limited. The only thing it provides is the links between each step in a sequential task plan and the 3D object involved. And several other factors important for real task planning are not provided. For example, how is each "step" decomposed into action (the action for agent to perform) and object (where the action is performed)? What are the pre- and post-conditions for each action? How can the agent assure that the desired object is really in the scene and how can it recover if it is missing? How to use tools (this paper assumes each step involves at most one object)?  Therefore, the proposed task and dataset are quite limited to the *3D object grounding (or navigation) with longer context* itself, and it does not seem to help much for real-world task planning.

2. It's not clear what's the embodiment of the proposed dataset, i.e. what agent is going to perform the task? Although the paper mentions "robot" several times, clearly some tasks/steps in the proposed dataset are not designed for robots, but for humans, like "enjoy reading before bed" and "wash your hands thoroughly in the sink". How can these instructions for humans be useful for AI? Later when it moves to the real-world execution and evaluation stage, how can the proposed dataset help task planning?

3. This paper does not propose a novel method (a stronger baseline) for the proposed task.

4. Images in Figure 9 Task 1 are too blurry. I cannot tell whether predictions are reasonable or not.

**Questions:**

Please see "weakness" section above.

---

> ### Author Response · Authors · 2024-11-21
> **Response to Reviewer 3oG2**
>
> We would like to thank the reviewer for the thoughtful evaluation of our manuscript. We appreciate the constructive feedback provided and will address each of the mentioned limitations point by point.
>
> > **W1:** The definition of the sequential grounding task is limited. The only thing it provides is the links between each step in a sequential task plan and the 3D object involved. And several other factors important for real task planning are not provided.
>
> We acknowledge that our sequential grounding task does not encompass the full process of task planning and execution. However, we believe that 3D object grounding represents a crucial intermediate step. In real-world task planning, an agent must first **locate** the position it needs to reach or the object it needs to interact with before executing any actions. Research on grounding objects in 3D scenes can enhance an agent's ability to interpret its surroundings and accurately identify target objects, contributing to more effective overall task planning.
>
> > **W2.1:** It's not clear what's the embodiment of the proposed dataset, i.e. what agent is going to perform the task?
>
> Currently, our setting does not involve a real robotic agent performing the tasks.
>
> > **W2.2:** Although the paper mentions "robot" several times, clearly some tasks/steps in the proposed dataset are not designed for robots, but for humans, like "enjoy reading before bed" and "wash your hands thoroughly in the sink". How can these instructions for humans be useful for AI?
>
> We acknowledge that some tasks in the proposed dataset are more natural for humans than for robots. However, these tasks hold value for AI in two key ways:
>
> 1. **Enhancing human-robot interaction:** For robots to effectively communicate and interact with humans, they need to understand a wide range of human daily activities—not just tasks typically designed for robots. This aligns with prior work like VirtualHome [1], which compiled diverse human activities (e.g., "brush teeth" and "go to sleep") to teach robots how to perform everyday tasks. More details are available [here](http://virtual-home.org/tools/explore.html).
> 2. **Applications for virtual characters:** These tasks can also be leveraged to instruct virtual characters in domains such as augmented reality (AR).
>
> [1] VirtualHome: Simulating Household Activities via Programs, CVPR 2018.
>
> > **W2.3:** Later when it moves to the real-world execution and evaluation stage, how can the proposed dataset help task planning?
>
> An embodied agent must understand its 3D environments and perform tasks specified by human users (like *"make me a cup of coffee"*). This requires the agent to **locate** the objects it needs to interact with (e.g., the *"coffee machine"*) in the scene to complete the task. Our work focuses on grounding task-related objects during task planning, a critical preparatory step before executing each action.
>
> > **W3:** This paper does not propose a novel method (a stronger baseline) for the proposed task.
>
> We appreciate the reviewer's concern regarding the lack of a novel method as a stronger baseline. However, the primary goal of this work is to introduce and establish a new benchmark. Developing methods to enhance model performance is a critical direction we plan to explore in future work.
>
> While we have not proposed a new baseline, we adapted several state-of-the-art models to evaluate their performance on our task. Notably, we modified the 3D LLM LEO architecture to enable grounding----an ability it didn't have originally. Despite the overall low performance of the baselines, 3D LLM LEO demonstrated superior results, particularly in task accuracy (t-acc). This finding underscores the potential of 3D LLMs in 3D visual grounding and provides valuable insights for advancing research in this area.
>
> > **W4:** Images in Figure 9 Task 1 are too blurry. I cannot tell whether predictions are reasonable or not.
>
> Thank you for pointing this out. We will replace the images in Figure 9 Task 1 with the higher-resolution version in our revised paper.

---

> > ### Comment · Reviewer_3oG2 · 2024-11-22
> >
> > I thank the authors for their comprehensive response.
> >
> > However, my concern about the limitations of the definition of the proposed task persists. I still think the significance of the sequential grounding task, in its current form, is limited and the contribution is a bit incremental compared to previous visual grounding benchmarks, as mentioned by current reviewers.
> >
> > I appreciate the systematic efforts in this work. I like the authors' response to reviewer CNtV, which clearly demonstrates the importance of the context and addresses some of my concerns.
> >
> > Therefore I tend to keep my rating by now.

---

> > > ### Author Response · Authors · 2024-11-27
> > > **Acknowledgement**
> > >
> > > Thank you for thoroughly reviewing our rebuttal. We are pleased that our response to Reviewer CNtV addressed some of your concerns, and we deeply value your feedback on improving the sequential grounding task's design to enhance its significance.

---

### Official Review · Reviewer_Nxxi · 2024-11-02

**Soundness:** 2
**Presentation:** 3
**Contribution:** 3
**Rating:** 5
**Confidence:** 5

**Summary:**

- The authors proposed a novel task, Task-oriented Sequential Grounding in 3D scenes, to address the gap between object-centric and task-driven grounding required for practical Embodied AI applications.
- The authors constructed a novel large-scale dataset for the Task-oriented Sequential Grounding in 3D scenes named SG3D, which contains 22,346 tasks with 112,236 steps across 4,895 real-world 3D scenes. They used GPT-4 based labeling of 3D-scenes from popular datasets: ScanNet, ARKitScenes, 3RScan, MultiScan, HM3D.
- The authors explored several representative approaches for solving the proposed task: the dual-stream multimodal model 3D-VisTA, the query-based model PQ3D, the 3D LLM LEO.
- The authors also made a working demonstration of the developed dataset available in an anonymous online project.

**Strengths:**

1. The authors created a large-scale SG3D visual grounding benchmark based on adding additional automated text markup to 3D scenes from well-known datasets ScanNet, ARKitScenes, 3RScan, MultiScan, HM3D. Such a dataset is useful for planning actions by an intelligent agent on static scenes.
2. The authors clearly demonstrated the advantages of the task-oriented steps in SG3D and object-centric referrals they created, including using the example of markup from ScanRefer for the same target objects.

**Weaknesses:**

1. The introduction and abstract note the usefulness of the created dataset for intelligent agents and robots, but the task examples shown in Figure 1 look strange for a robot agent. They may be useful for NPC characters in computer games, but the authors do not mention this in the introduction. In this regard, the authors are asked to adjust the introduction to more accurately explain the motivation for the work done.

2. To demonstrate the quality of modern methods on the created dataset, the authors use only three models 3D-VisTA, PQ3D, and LEO (one model per category: Dual-stream, Query-based, 3D LLM). At the same time, their choice is not sufficiently justified. In Related Works, the authors also note the existence of other models that are not included in the comparison of approaches, which reduces the quality of the constructed benchmark. This also raises questions about the completeness and sufficiency of the results shown in Table 1. The Appendix section contains metrics for three other models: Vil3DRef, ViewRefer, MiKASA-3DVG, it is recommended to move them to the main text of the article, but the choice still requires explicit justification.

3. The article lacks an ablation study for the selected modifications of the 3D-VisTA, PQ3D, and LEO models, which would also add validity to the results obtained. Moreover, the adaptation of these models is noted as one of the contribution points.

4. During the benchmark, the authors considered the combination of only pre-trained GPT-4 (with closed code) and 3D object classifier. However, it would be interesting to see the same experiment with other open-source models, the results of which would be easier for researchers to reproduce.

**Questions:**

1 Why was GPT-4 chosen to generate diverse tasks when creating the dataset? Did the authors try using other LLMs (Mistral/LLAMA) for this?

---

> ### Author Response · Authors · 2024-11-21
> **Response to Reviewer Nxxi [1/2]**
>
> We would like to thank the reviewer for the thoughtful evaluation of our manuscript. We appreciate the constructive feedback provided and will address each of the mentioned limitations point by point.
>
> > **W1:** The introduction and abstract note the usefulness of the created dataset for intelligent agents and robots, but the task examples shown in Figure 1 look strange for a robot agent. They may be useful for NPC characters in computer games, but the authors do not mention this in the introduction. In this regard, the authors are asked to adjust the introduction to more accurately explain the motivation for the work done.
>
> Thank you for your suggestion. We will revise the introduction to more accurately convey the motivation behind our work, ensuring it aligns with the potential applications for both intelligent agents and virtual characters.
>
> > **W2:** To demonstrate the quality of modern methods on the created dataset, the authors use only three models 3D-VisTA, PQ3D, and LEO (one model per category: Dual-stream, Query-based, 3D LLM). At the same time, their choice is not sufficiently justified. In Related Works, the authors also note the existence of other models that are not included in the comparison of approaches, which reduces the quality of the constructed benchmark.
>
> We thank the reviewer for their insightful feedback. Evaluating representative models on a newly proposed benchmark is a standard approach to highlighting its attributes, as demonstrated in Multi3DRefer [1]. Our selection of baselines was carefully designed to represent a broad range of model architectures in the 3D vision-language (3D-VL) field.
>
> We chose three representative model types----dual-stream, query-based, and multimodal LLM—following established practices in multimodal research. Specifically, 3D-VisTA, PQ3D, and LEO were selected as the most representative and state-of-the-art (SOTA) models in their respective categories, compared to other approaches discussed in the Related Work section.
>
> Our experiments revealed two key findings: (1) existing models struggle significantly on the SG3D task, and (2) 3D LLMs exhibit markedly higher potential compared to other types of 3D-VL models. While we also benchmarked three additional models (Vil3DRef, ViewRefer, and MiKASA-3DVG), they did not provide additional insights.
>
> To address the reviewer's concern, we will move the metrics for these additional models from the Appendix to the main text in future revisions. Additionally, we are committed to including more baselines in subsequent work to strengthen the robustness and comprehensiveness of our benchmark.
>
> [1] Multi3DRefer: Grounding Text Description to Multiple 3D Objects (ICCV 2023)
>
> > **W3:** The article lacks an ablation study for the selected modifications of the 3D-VisTA, PQ3D, and LEO models, which would also add validity to the results obtained. Moreover, the adaptation of these models is noted as one of the contribution points.
>
> 3D-VisTA and PQ3D inherently support grounding, so we did not alter their original configurations. However, we modified the 3D LLM LEO architecture to incorporate grounding, a capability it initially lacked. We conducted two ablation experiments on these modifications to LEO's structure, and the results will be shared soon.
>
> > **W4:** During the benchmark, the authors considered the combination of only pre-trained GPT-4 (with closed code) and 3D object classifier. However, it would be interesting to see the same experiment with other open-source models, the results of which would be easier for researchers to reproduce.
>
> We evaluate the combination of the open-source LLM *qwen2-72b-instruct* and 3D object classifier, with results summarized below:
> |                    | ScanNet s-acc      | ScanNet t-acc      | 3RScan s-acc       | 3RScan t-acc       | MultiScan s-acc    | MultiScan t-acc    |   ARKitScenes s-acc   | ARKitScenes t-acc   | HM3D s-acc          | HM3D t-acc          | OverAll s-acc       | OverAll t-acc       |
> |--------------------|---------------------|---------------------|---------------------|---------------------|---------------------|---------------------|--------------------|--------------------|---------------------|---------------------|---------------------|---------------------|
> | qwen2-72b-instruct | 44.2                | 12.5                | 26.5                |  2.4                 | 22.2                |  0.0                |  25.2                |  6.0                | 20.2                |  4.8                 | 26.8                |  6.3                |
>
> This approach achieved an s-acc of 26.8% and a t-acc of 6.3%, which is slightly lower than the "GPT-4 with a 3D object labeler" baseline and significantly underperforms fine-tuned 3D-VL models. These results further support our conclusion in Section 5.2 that fine-tuned 3D-VL models are better suited for this task.

---

> > ### Author Response · Authors · 2024-11-21
> > **Response to Reviewer Nxxi [2/2]**
> >
> > > **Q1:** Why was GPT-4 chosen to generate diverse tasks when creating the dataset? Did the authors try using other LLMs (Mistral/LLAMA) for this?
> >
> > We chose GPT-4 for its state-of-the-art creativity, and stability. Although not entirely free from hallucinations or flawed steps, it produces fewer errors than other models like GPT-3.5, making it the most reliable choice for generating tasks from long scene graphs.

---

> > > ### Author Response · Authors · 2024-11-25
> > > **Ablation study on modifications of 3D LLM LEO**
> > >
> > > > **W3:** The article lacks an ablation study for the selected modifications of the 3D-VisTA, PQ3D, and LEO models, which would also add validity to the results obtained. Moreover, the adaptation of these models is noted as one of the contribution points.
> > >
> > > As previously mentioned, we conducted two ablation experiments to analyze the structural modifications to LEO. These experiments were trained for 50 epochs on the *ScanNet* dataset with all other settings identical to the main experiments. Below are the details:
> > >
> > > ### **1. Source of object tokens in grounding**
> > > In LEO, the grounding head inputs the concatenation of all object tokens and the grounding token [GRD] to identify the target object. We tested two sources for the object tokens:
> > > - **Pre-tokens**: Object tokens generated before passing through the LLM (final choice in our model).
> > > - **Post-tokens**: Object tokens derived from the LLM's last hidden state.
> > >
> > > The performance under each setting is summarized below:
> > >
> > > | Model | t-acc (%) | s-acc (%) |
> > > |---|---|---|
> > > | LEO (pre-tokens) | **22.1** | **58.2** |
> > > | LEO (post-tokens) | 0.7 | 11.6 |
> > >
> > > The results indicate that the "post-tokens" setting performs significantly worse. This can be attributed to the grounding token [GRD] requiring object tokens with pure semantic information. Tokens from the LLM's last hidden state (post-tokens) likely contain additional context or irrelevant information, which hampers grounding accuracy. Therefore, we selected "pre-tokens" as the source of object tokens.
> > >
> > > ### **2. Different grounding heads**
> > > We evaluated three grounding head designs for LEO:
> > >
> > > - **MLP grounding head** (final choice in our model): Combines object and grounding embeddings through concatenation, followed by an MLP to predict grounding logits. It is lightweight and directly integrates features for grounding.
> > > - **Contrastive grounding head**: Maps object and grounding embeddings into a shared latent space via MLPs. Grounding logits are computed as the dot product in this space, focusing on embedding alignment.
> > > - **Self-attention grounding head**: Employs a transformer encoder to fuse object and grounding embeddings through multi-head self-attention, enabling complex interactions between the inputs.
> > >
> > > Performance comparisons are summarized below:
> > >
> > > | Model                     | t-acc (%) | s-acc (%) |
> > > |---------------------------|-----------|-----------|
> > > | LEO (MLP head)            | **22.1**      | **58.2**      |
> > > | LEO (contrastive head)    | 19.6      | 56.5      |
> > > | LEO (self-attention head) | 13.4      | 48.7      |
> > >
> > > The MLP head achieves the best performance, balancing simplicity and effectiveness. Thus, we finally select the MLP grounding head.
> > >
> > > We will include these results in our revised paper.

---

> > > > ### Comment · Reviewer_Nxxi · 2024-11-25
> > > >
> > > > I am grateful to the authors for the response and additional experiments they conducted.
> > > >
> > > > At the same time, my comments were not taken into account in the revised version of the article, which the authors could upload to the system, but were limited to replies to the reviewer. The authors' plans to change the article in the future are unclear, and I evaluate the materials that have been provided now.
> > > >
> > > > My main comments remain the same - the authors should have added more comparisons of their approach with existing state-of-the-art solutions. It is commendable that the authors conducted an experiment with the open-source qwen2-72b-instruct model, but there are other LLMs, why did they choose only this additional model?
> > > >
> > > > The authors' contribution "We adapted three state-of-the-art 3D visual grounding models (3D-VisTA, PQ3D, and LEO) to the sequential grounding task and evaluated them on SG3D." after the response to my comment becomes much weaker, since the authors adapted not three models, but only one (LEO).
> > > >
> > > > I will keep my rating (5) the same.

---

> > > > > ### Author Response · Authors · 2024-11-27
> > > > > **Acknowledgement**
> > > > >
> > > > > Thank you for taking the time to carefully review our rebuttal. We appreciate your valuable suggestion to include broader comparisons of both 3D-VL approaches and LLM-based approaches.

---

### Official Review · Reviewer_CNtV · 2024-11-03

**Soundness:** 2
**Presentation:** 3
**Contribution:** 2
**Rating:** 3
**Confidence:** 5

**Summary:**

This paper introduces a new visual grounding task called Task-oriented Sequential Grounding. It presents a large-scale dataset SG3D and its benchmark and makes efforts to move beyond traditional 3D visual grounding towards more practical, task-oriented applications. On this new benchmark, this paper implements different models to fit the sequential VG setting and analyze the new challenges with comprehensive experimental results.

**Strengths:**

1. It designs a Visual Grounding task more closely aligned with real-world applications.
2. The paper provides a detailed discussion of the process for generating the SG3D dataset, including the specifics of how prompts are designed to generate tasks and the measures taken to minimize potential biases during the generation process.
3. The paper provides the basic experimental results on the new benchmark and implements several baselines from different paradigms, and a human study further validates the benchmark's validity.

**Weaknesses:**

The main concern is that this benchmark does not fundamentally differ from traditional VG benchmarks, or at least does not justify what new significant challenges this benchmark brings (both in the data generation pipeline and experimental analysis). From my perspective, It simply incorporates a "task" narrative background into Visual Grounding, and the "Task-oriented" is reflected only in the (1) use of task information in samples and (2) sequential text inputs. However, how these two designs bring new fundamental problems to the VG problem needs further exploration. For example, the task-oriented setting can incorporate the situation information or more complex semantic understanding (wash hands -> find a sink) under this background, but this has also been discussed in Situated QA (SQA) and ScanReason. Or this setting involves navigation to make the setting different from pure VG / involves context understanding (among sequential actions) in a single-step solution.

In summary, the authors need to figure out how to better justify the fundamental value of this benchmark, showing it is not just a simple combination of several single-step VG with only task-related prompts. I strongly recommend that the author improve the paper and make this new benchmark more convincing. I would also consider raising the score if the author can provide a satisfied response or explanation regarding this concern.

**Questions:**

1. In this benchmark, could you provide examples of how context is demonstrated in its role? Is a model only required to have the ability to ground objects from the text of a single action step to complete this task? An ideal example would be when there are multiple instances with the same object type (distractors), where the model needs to use information from previous action steps to select the correct one among these, although this example may also be incremental compared to previous VG benchmarks (only longer context for VG). Are there such examples, and if they exist, can you provide and quantify the proportion of such cases in SG3D?
2. For the analysis of the effect of offering contextual information in the Ablation Study, to ensure consistency, I believe a reasonable approach is the following: Use the same model trained on the original training set and test it both on the original test set and a test set “with only the current action step,” then compare the results. Alternatively, train models separately on the original training set and a training set with “only the current action step” and test both models on the original test set to compare the experimental results.

---

> ### Author Response · Authors · 2024-11-20
> **Response to Reviewer CNtV [1/2]**
>
> We would like to thank the reviewer for the thoughtful evaluation of our manuscript. We appreciate the constructive feedback provided and will address each of the mentioned limitations point by point.
>
> > **W1:** The main concern is that this benchmark does not fundamentally differ from traditional VG benchmarks, or at least does not justify what new significant challenges this benchmark brings.
>
> We argue that SG3D introduces two significant challenges compared to traditional VG benchmarks:
>
> 1. **Implicit situation inference across steps:** Unlike SQA3D and ScanReason, which provide explicit situational contexts (e.g., “Sitting at the edge of the bed and facing the couch” in SQA3D), SG3D requires models to infer context implicitly from task descriptions and prior steps. For instance, in the *"Enjoying reading before bed"* task (Figure 1), step 2 (*"sit on the mattress on the floor"*) implies the reading location, guiding the model to select the nearest lamp (lamp-20) in step 3, rather than a farther lamp (lamp-19) at the other bedside.
> 2. **Referring consistency across steps:** SG3D emphasizes consistent object references throughout a sequence. For example, in the *"Make a cup of coffee and serve it on a plate"* task (Figure 1), step 6 (*"go back to the table and put down the coffee"*) requires identifying the same table used in step 3, unlike single-step benchmarks that do not address this consistency.
>
> > **Q1.1:** In this benchmark, could you provide examples of how context is demonstrated in its role?
> > An ideal example would be when there are multiple instances with the same object type (distractors), where the model needs to use information from previous action steps to select the correct one among these
>
> We appreciate the reviewer’s insightful question. Some tasks in Figure 1 indeed include such examples, though the small image sizes might make it challenging to fully illustrate how context plays a role without further explanation. Below, we highlight two specific examples from Figure 1 that demonstrate the model's reliance on prior steps to resolve distractors:
>
> 1. **Task: Enjoying reading before bed**  (top-left in Figure 1)
> Step 3 requires the model to use information from step 2.
> - **Scene information** Two bedside lamps are present—lamp-20 on the left and lamp-19 on the right.
> - **Step 2:** *"Sit on the mattress on the floor"* specifies the position as the mattress at the left bedside.
> - **Step 3:** *"Turn on the lamp to provide light"* requires distinguishing between the two lamps. Context from step 2 clarifies that the target is lamp-20 on the left, next to the mattress.
> 2. **Task: Refresh yourself with a beverage** (bottom-left in Figure 1)
> Step 3 requires the model to use information from step 1.
> - **Scene information:** Two desks are visible—a longer desk (desk-10) on the left and a shorter desk (desk-9) on the right. Each desk has a black office chair underneath (chair-12 under desk-10, chair-13 under desk-9).
> - **Step 1:** *"Walk to the shorter one of two desks with a monitor"* identifies desk-9 (the right desk) as the target.
> - **Step 3:** *"Sit on the black office chair under that same desk to enjoy your drink"* links "that same desk" to desk-9, requiring the model to identify chair-13 under desk-9, excluding chair-12 under desk-10.

---

> > ### Author Response · Authors · 2024-11-20
> > **Response to Reviewer CNtV [2/2]**
> >
> > > **Q1.2:** Is a model only required to have the ability to ground objects from the text of a single action step to complete this task?
> >
> > No.
> >
> > Our empirical analysis highlights the necessity of contextual information through an ablation study. We trained models on the original training set and evaluated them on two test sets: the original test set (with context) and a modified version containing only the current action step (without context). The results below illustrate a significant performance drop when context is removed, underscoring the importance of grounding objects based on both the current step and prior context:
> > |                  | 3D-VisTA | PQ3D  | LEO   |
> > |------------------------|----------|-------|-------|
> > | t-acc w/ context (%)   | 30.6     | 26.8  | 34.1  |
> > | t-acc w/o context (%) | 16.7     | 17.2  | 22.3  |
> > | $\Delta$ (%)       | -13.9    | -9.6  | -11.8 |
> >
> > Additionally, we examined cases where the 3D LLM LEO succeeded with context but failed without it. These failures often stem from **referring consistency**, where later steps reference objects from prior steps using pronouns (e.g., "it," "the other," "here") or simplified expressions (e.g., "the XXX"). This omission mimics human communication patterns and creates challenges without prior context. Below are two examples:
> > ```
> > Task: Organize documents into the cabinet.
> > 1. Retrieve documents from the green desk chair.
> > 2. Walk to the white, metal filing cabinet with drawers.
> > 3. Store the documents in one of the drawers of the cabinet.
> >
> > Ground truth:
> > 1. [chair-10]
> > 2. [cabinet-12]
> > 3. [cabinet-12]
> >
> > prediction with context (correct):
> > 1. [chair-10]
> > 2. [cabinet-12]
> > 3. [cabinet-12]
> >
> > prediction without context:
> > 1. [chair-10]
> > 2. [cabinet-12]
> > 3. [cabinet-11] (incorrect; inconsistent with step 2)
> > ```
> >
> > ```
> > Task: Prepare a relaxing bath.
> > 1. Go to the bathtub resting on the shower floor.
> > 2. Turn on the faucet inside the bathtub.
> > 3. Adjust the water temperature to a comfortable level using the same faucet.
> > 4. Place the potted plant from the top of the bathtub to the right side of the bathroom vanity.
> > 5. Wait for the bathtub to fill with warm water.
> > 6. Turn off the faucet when the bathtub is full.
> >
> > Ground truth:
> > 1. [bathtub-96]
> > 2. [faucet-153]
> > 3. [faucet-153]
> > 4. [potted plant-69]
> > 5. [bathtub-96]
> > 6. [faucet-153]
> >
> > prediction with context (correct):
> > 1. [bathtub-96]
> > 2. [faucet-153]
> > 3. [faucet-153]
> > 4. [potted plant-69]
> > 5. [bathtub-96]
> > 6. [faucet-153]
> >
> > prediction without context:
> > 1. [bathtub-96]
> > 2. [faucet-153]
> > 3. [faucet-110] (incorrect; faucet-110 belongs to the sink, inconsistent with step 2)
> > 4. [potted plant-69]
> > 5. [bathtub-96]
> > 6. [faucet-153]
> > ```
> >
> > > **Q1.3:** Can you quantify the proportion of such cases in SG3D?
> >
> > We estimate the proportion of context-critical cases based on our ablation study results. Among the tasks where LEO succeeds with context, 46.4% are context-critical, meaning LEO fails these tasks without context. The remaining 53.6% are context-independent, where LEO succeeds regardless of whether context is provided.
> >
> > > **Q2:** For the analysis of the effect of offering contextual information in the Ablation Study, to ensure consistency, I believe a reasonable approach is the following: Use the same model trained on the original training set and test it both on the original test set and a test set “with only the current action step,” then compare the results.
> >
> > We appreciate the reviewer’s suggestion regarding the ablation study on contextual information. Following this approach, we trained each model on the original training set and evaluated it on two test sets: the original test set (with context) and a modified test set containing only the current action step (without context).
> >
> > The results show significant performance drops in task accuracy when context is removed:
> > |                  | 3D-VisTA | PQ3D  | LEO   |
> > |------------------------|----------|-------|-------|
> > | t-acc w/ context (%)   | 30.6     | 26.8  | 34.1  |
> > | t-acc w/o context (%) | 16.7     | 17.2  | 22.3  |
> > | $\Delta$ (%)       | -13.9    | -9.6  | -11.8 |
> >
> > We will include these findings in the revised version of our paper.

---

> ### Comment · Reviewer_CNtV · 2024-11-22
> **Decision after Rebuttal**
>
> Thanks for the feedback from the authors. The statistics and ablation studies about the role of context in the samples of this benchmark are interesting and address some of my concerns. However, the setting of this task, including the same scene input for each step, the output that only contains the target's ID, and the model that keeps the conventional input/output design, makes the current paper with the current state unable to fully reflect the motivation. Therefore, I encourage the authors to polish the benchmark and methods further to better highlight the challenges in this task, i.e., long context and consistent grounding results during the procedure from the rebuttal, making the paper's study fundamentally different from previous grounding tasks.
> Given these concerns and other reviews, I would keep my original rating (although it tends to be an upgraded score of 4).

---

> > ### Author Response · Authors · 2024-11-27
> > **Acknowledgement**
> >
> > Thank you for taking the time to carefully review our rebuttal. We are glad that our clarifications on the role of context addressed some of your concerns. We truly appreciate your encouragement to refine the manuscript further, and we are committed to enhancing the benchmark and methods to better align with our motivation and emphasize the new challenges of this task.

---

### Meta-Review · Area_Chair_pbGy · 2024-12-21

**Metareview:**

The paper introduces Task-oriented Sequential Grounding in 3D scenes, where an agent must follow multi-step instructions to locate a sequence of target objects. Unlike traditional 3D visual grounding benchmarks that focus on single-step, object-centric queries, this new task emphasizes long-horizon context and the interplay between consecutive steps. To support this, the authors contribute SG3D, a large-scale dataset of 22,346 tasks and 112,236 steps across 4,895 3D indoor scenes drawn from several well-known 3D scene sources (e.g., ScanNet, ARKitScenes, 3RScan, MultiScan, HM3D).

Strengths of the Paper:

-- The proposed sequential grounding task more realistically reflects how instructions and object references appear in practical applications (e.g., daily tasks, household activities).

-- SG3D is systematically constructed and includes human verification, spanning a variety of 3D scenes with detailed step-by-step instructions, thereby demonstrating considerable effort in data gathering and quality assurance.

-- Adapting multiple existing 3D vision-language models (e.g., 3D-VisTA, PQ3D, LEO) to the sequential grounding setting offers a concrete performance baseline and reveals where current approaches fall short.

Weaknesses of the Paper

-- While the dataset is labeled task-oriented, some reviewers feel that the “task” element is mostly reflected as extra narrative and sequential text rather than introducing fundamentally novel technical challenges (e.g., navigation, memory, real-world planning).

-- The paper does not fully demonstrate how the multi-step context leads to substantially different challenges compared to traditional single-step 3D visual grounding benchmarks, beyond merely providing longer text inputs.

-- The paper lacks thorough ablation studies and deeper comparisons with more models in the main text, making it less clear which adaptations are most critical or how different design choices impact performance on SG3D.

While the work opens an important direction for visual grounding in 3D, it leaves some open questions about the truly task-oriented nature of these sequences and how they depart from a series of single-step queries.  After carefully reading the paper, the reviews, and rebuttal discussions, the AC agrees with the reviewers on recommending to reject the paper.

**Additional Comments On Reviewer Discussion:**

The strengths and weaknesses are described above.

---

### Decision · Program_Chairs · 2025-01-22

Reject